# Role of Spanish Toddlers’ Education and Care Institutions in Achieving Physical Activity Recommendations in the COVID-19 Era: A Cross-Sectional Study

**DOI:** 10.3390/children9010051

**Published:** 2022-01-03

**Authors:** Herminia Vega-Perona, Isaac Estevan, Yolanda Cabrera García-Ochoa, Daniel A. Martínez-Bello, María del Mar Bernabé-Villodre, Vladimir E. Martínez-Bello

**Affiliations:** 1Department of Teaching of Musical, Visual and Corporal Expression, University of Valencia, 46022 Valencia, Spain; hervepe@alumni.uv.es (H.V.-P.); isaac.estevan@uv.es (I.E.); maria.mar.bernabe@uv.es (M.d.M.B.-V.); 2COS Research Group, Body, Movement, Music and Curricular Practices, University of Valencia, 46022 Valencia, Spain; yolanda.cacbrera@uv.es; 3AFIPS Research Group, University of Valencia, 46022 Valencia, Spain; 4Department of Language Theory and Communication Sciences, Faculty of Philology, Translation and Communication, University of Valencia, 46010 Valencia, Spain; 5Programa de Bacteriología y Laboratorio Clínico, Facultad de Ciencias Médicas y de la Salud, Universidad de Santander, Bucaramanga Cl. 45 11-52, Colombia; danieladyro@gmail.com

**Keywords:** physical activity, early childhood education and care, toddlers, COVID-19, schools

## Abstract

To our knowledge, there are no published studies that describe the physical activity (PA) levels and objectively measure them through accelerometry in toddlers (2–3 years old) attending early childhood education and care (ECEC) institutions during the COVID-19 pandemic. The aims of this study were two-fold: (a) to analyse toddlers’ PA levels and sedentary behaviour (SB) during school hours in ECEC institutions, as well as the rate of adherence to specific recommendations on total PA (TPA) and moderate–vigorous PA (MVPA); and (b) to evaluate the characteristics correlates (age, gender, and body mass index –BMI) of young children and the school environment on toddlers’ TPA, light PA (LPA), MVPA, and SB during school hours in ECEC institutions. PA was evaluated with ActiGraph accelerometers. The main findings were that: (a) toddlers engaged in very high amounts of TPA and MVPA during ECEC hours; (b) girls and boys displayed similar levels of LPA, TPA, and SB, while girls had lower levels of MVPA, compared to boys, and younger toddlers were less active than older ones; (c) BMI was not associated with PA of any intensity or SB; (d) playground and classroom density were not associated with higher levels of PA of any intensity, though classroom density was associated with SB. These ECEC institutions provide and challenge the new COVID-19 scenario, as well as supportive environments for toddlers’ PA.

## 1. Introduction

Between January 2020, when the World Health Organization (WHO) declared a new coronavirus disease 2019 (COVID-19) outbreak to be a public health emergency of international concern [1]; at the time of writing (August 2021), there have been more than 200 million confirmed infections and over 4 million human deaths [2]. To mitigate the spread and infection, governments from around the world have adopted measures to limit social contact, including stay-at-home orders [3], travel restrictions, and school closures [4]; upon reopening, both the schools and their students and families have had to adapt to a new pandemic reality [4,5]. These extraordinary arrangements have had negative social, emotional, and physical impacts on young children, including sleeping disorders, reduced physical activity (PA), and increased sedentary behaviour (SB) [6,7,8,9]. When space to play has been provided, the increased SB and decreased PA in toddlers have been attenuated [10].

Regarding PA in toddlers, international PA guidelines recommend 180 min/day of total PA (TPA: light, moderate, and vigorous) and no more than one hour sedentary screen time [11], progressing toward 60 min of moderate–vigorous PA (MVPA) by age five. Other institutions suggest that toddlers should have 30 min of structured PA, in addition to 60 min of unstructured PA per day [12]. According to the WHO [11], adherence to these recommendations supports cognitive and motor development in toddlers. So, early childhood education and care (ECEC) institutions must provide opportunities for them to participate in PA for at least 15 min of active play per hour [13] because active play contributes to their cognitive, physical, social, and emotional well-being [14].

Multiple correlates of PA, explained by ecological models of health behaviours [15], have postulated the existence of different systems impacting children’s behaviours at individual and environmental levels. At the individual level, studies in toddlers have focused on analysing PA correlates, such as age, gender, and body mass index (BMI). Whereas toddlers’ BMI has not been shown to be associated with PA [16,17], being a boy is associated with high levels of PA [16,18,19,20], which intensifies as they grow older [16,21,22]. Such individual correlations during school hours were evident even before the COVID-19 pandemic, wherein older toddler boys accumulated more active behaviours than girls and younger children [23,24].

At the environmental level, Hunter et al. [25] suggested that proximal factors may be key predictors of toddlers’ PA and SB. As young children spend a high percentage of time in ECEC institutions, and these settings have an important role in promoting healthy habits [26]; it is essential to understand how ECEC environmental and individual factors influence toddlers’ movement behaviours. Environmental factors can include program structure [27], characteristics of the indoor environment (such as the ‘modified open-plan space’) [28], the presence of fixed and portable equipment [29], modifiable practices (such as routines), increased time spent in outdoor environments, and certain features of the outdoor environment [30]. Individual factors encompass the child care educator modelling or the co-participation of sedentary time and PA [31]. In fact, Lahuerta-Contell et al. [32] found that a high density of preschoolers was associated with higher levels of SB during school hours. Thus, given the multidimensional correlates of children’s PA, ECEC stand out as an important setting for young children’s development, providing safe and convenient opportunities for children to be active [33].

On the other hand, the pandemic has driven schools to make adaptations to their internal operations, creating measures for prevention, hygiene, and health promotion in the face of COVID-19 [5]. This new scenario has impacted movement behaviours for children and entailed new challenges for teachers, including the improvement of the methodological competences related to conducting PA classes that are attractive and adapted to the conditions during the pandemic, as well as training educators to motivate pupils to be more physically active [34,35]. For instance, Lafave et al. [35] found that measures implemented in ECEC centres to protect young children from COVID-19 have also limited teachers’ perceived ability to promote PA by limiting PA space, constraining access to equipment, and reducing the variety of activities on offer.

Given the major challenges to leisure-time PA that young children and adults have been facing in the COVID-19 era [10,36], ECEC institutions have a responsibility to minimise, insofar as they can, these impacts on PA levels [35,37]. To date, there have been no published studies that describe objectively measured the PA levels in toddlers (2–3 years old) attending ECEC institutions during the COVID-19 pandemic. It is possible that this pandemic time could challenge prevailing assumptions about both the correlates of young children’s movement behaviour and how teachers and practitioners take a stand in promoting and offering quality curricular PA [38]. A recent systematic review [39] found that PA estimates are inconclusive and largely heterogeneous and, despite the fact that toddlers tend to exceed the TPA recommendation of 180 min/day, very little of this time is spent at higher movement intensities. Furthermore, although the ECEC context is an important setting for PA promotion, relatively few studies focus on toddlers’ PA and SB in ECEC hours [27,28,29,30]. Therefore, the aims of this study were two-fold: (a) to analyse toddlers’ PA levels and SB during school hours in ECEC institutions during the COVID-19 pandemic, as well as the rate of adherence to specific recommendations on TPA and MVPA; and (b) to evaluate the child’s characteristics correlates (child’s age, gender, and BMI) of young children and the school environment on toddlers’ TPA, light PA (LPA), MVPA, and SB during school hours in ECEC institutions.

## 2. Materials and Methods

### 2.1. Design and Participants

The study used a cross-sectional, correlational design. A total of 180 young children, aged 24 to 36 months from seven different public ECEC institutions in metropolitan Valencia, Spain, were recruited by way of convenience sampling after a formal invitation to participate in the study. Inclusion criteria were: children aged 2–3 years; being able to walk without assistance; and attending the ECEC institution at least three days per week. Young children whose parents signed consent, who attended an ECEC, and who were developmentally typical were eligible to participate in the study. Young children (n = 60) who did not attend the school for the five consecutive days of data collection or went home early (e.g., due to illness or lack of attendance at the school when measurements were carried out) were excluded from the final analyses.

After applying these criteria, the final sample was made up of 120 toddlers (mean age 2.5 years, standard deviation [SD] 0.5; 45.8% girls; Table 1). The parents or guardians of all children signed informed consent before study commencement. The Human Research Ethics Committee, at the corresponding author’s university, approved the study protocol (ethical approval code- INV_ETICA-1441131). Data were collected from October 2020 to May 2021. After parents/guardians of participating children had signed the consent form, they completed a demographic questionnaire during the same week of the PA measurement with accelerometers. This questionnaire elicited data on toddlers’ gender, age, weight, height, ECE enrolment status, and several family-related variables (e.g., parent/guardian completing the questionnaire, relationship with the young children, etc.). BMI was calculated and categorised in accordance with gender- and age-based criteria as follows: underweight; healthy weight; overweight; and obese [40]. With regard to the BMI data, we used the information provided by the parents for a few reasons. First, the study period coincided with the third wave of the spread of the COVID-19 virus in our country, so the restrictions for accessing the centre were quite strict. Secondly, as we could not enter the centres ourselves (only drop off questionnaires and accelerometers), we could not directly determine BMI, as this would have violated social distancing rules. For these reasons, we opted to have the parents provide children’s height and weight, as they could measure these parameters without restrictions. Moreover, at these ages (24 to 36 months), visits and check-ups with paediatricians are quite common, and the data are easily accessible to families.

### 2.2. ECEC Background Information and School Hours

The research group invited seven ECEC institutions from five towns in the metropolitan Valencia area to participate in the study. All accepted the invitation. Two ECECs were from one city, and each of the other four cities had one participating ECEC. All ECECs were located in socioeconomically similar areas. Background information regarding the ECEC institutions, such as the number of enrolled children, was recorded during interviews with the principal of each ECEC institution. The centres were public (local and/or regional government-owned) and offered full-time instruction and care. In all the ECEC institutions, there was a recess time, lunchtime, and nap time, lasting a mean of 45 min, 30 min, and 90 min, respectively. In all the ECEC institutions, lunch was served in the same building, while nap time was in the toddlers’ classroom. Two educators were present in each toddler classroom. In addition, during the week of the measurements, the teachers responsible for each classroom were asked about the number of children attending regularly. Of the seven participating ECEC institutions, three had specific structured PA sessions within their curricular practices (in the weekly schedule and in a special classroom), while the other four worked PA curricular contents in an integrated, interdisciplinary way. All ECECs had fixed equipment in their outdoor playgrounds.

### 2.3. ECEC and COVID-19 Regulations

The rules of social distancing, hygiene, and cleanliness, as well as other logistical and institutional rules for both teachers and children remained in place from the beginning of the school year (September 2020) to the end (June 2021). These policies were common for all the ECEC institutions. Following governmental regulations for maintaining a safe distance and avoiding contact between different groups, one of the restrictions imposed to prevent the spread of COVID was the creation of stable classroom groups that do not socialise with children outside their group. Furthermore, school principals were asked how the different groups were organised during recess time to comply with the new COVID-19 regulations, including what modifications had been made to the play area and how many children per space participated during playtime.

### 2.4. Objective Physical Activity Measurement

PA levels were measured with ActiGraph accelerometers (GT3X+Actigraph, Pensacola, FL, USA), for which, the use has been validated to assess PA in toddlers [41,42]. For five consecutive weekdays, toddlers wore the accelerometer on their right hip from the time of their arrival at the ECEC institutions until they left for the day. The ECEC teachers responsible for each classroom placed the accelerometer on the children at the beginning of the school day (9:00–9:15 am) and removed it a few minutes before the end (4:15–4:30 pm). Nap time was considered non-wear time and excluded from the analysis, as over 90% of children this age still nap [24]. Mean daily wear-time was 7.3 h; after excluding lunch and nap time, mean daily wear-time for PA and SB measurements was 5.3 h). After the accelerometers were collected, data were downloaded in 15-s epochs to capture the sporadic and intermittent activity pattern of the young children [41,42]. Teachers were given a record sheet for noting any comments regarding their use during the study. PA was scored using Trost’s cutoffs for 2-year-old children: the 15-s count range corresponding to SB was 0–48 counts/15 s; for LPA, 49–418 counts/15 s; and for MVPA, >418 counts/15 s, [43]. TPA was defined as any activity of light to vigorous intensity (LPA+MVPA) [42].

### 2.5. Classroom and Playground Densities

The indoor classroom and indoor playground area (m^2^) in each ECEC institution were measured manually with a meter. Classroom sizes ranged from 35 m^2^ to 42 m^2^. For the outdoor playground area, Google Earth Pro (GEP) software was used to provide an estimate of the playground spatial area (m^2^) at each of the ECEC institutions using aerial pictures of the playgrounds and the polygon measurement tool. Playground sizes ranged from 67 m^2^ to 240 m^2^). Classroom and playground densities (toddlers/m^2^) were calculated as described elsewhere [44]: average classroom density was calculated by dividing the number of toddlers in the classroom by the classroom size available for use indoor time. Average playground density was calculated by dividing the number of toddlers in the playground by the playground size available for use outdoor time.

### 2.6. Statistical Analysis

Descriptive statistics were used to calculate means and SDs for the continuous variables. Mixed-effects linear models were applied to establish the association between LPA, MVPA, TPA, and SB as outcome variables, as well as the independent variables age, gender, classroom density, and playground density as fixed effects, including classrooms as random effects with a normal distribution centred at 0 and variance for the classroom effects. BMI was included in initial models, but it did not show significant associations with the outcome variables. In addition, mixed-effects models with random effects for schools were not associated with the outcomes. For decision-making, a 5% significance level was applied. R software, version 4.0, was used for the computation, based on the lme4 package, version 1.1-26.

## 3. Results

The average time spent in TPA per day was 154.35 (SD 30.60) minutes, comprising a mean 111.85 ± 21.78 min of LPA and 42.50 ± 17.50 min of MVPA. The mean duration of SB was 164.47 ± 33.85 min. Figure 1 presents the mean time (min) spent in each PA category during school hours by gender. After adjusting for gender and age, boys engaged in significantly more minutes of MVPA than girls 46.91 ± 19.63 min vs. 37.28 ± 12.92 min, *p* < 0.05. For other PA outcomes, results were similar in boys and girls: LPA, 110.80 ± 19.76 min vs. 113.09 ± 24.05 min; TPA, 157.70 ± 31.41 min vs. 150.37 ± 29.41 min; and SB, 166.39 ± 37.1 vs. 162.19 ± 29.71 min, respectively.

Regarding the size of the ECEC, the average classroom was 38.06 m^2^ (range 35 to 42), and the playground area, 157.58 m^2^ (range 67 to 240). The classroom and playground densities, measured as the average number of young children per square meter of classroom and playground space, were 2.78 children/m^2^ (range 2.30 to 3.5) and 11.36 children/m^2^ (range 4.61 to 16), respectively.

Table 2 shows the results of mixed-model regressions, after adjusting for gender, age, and BMI. There were significant associations between objectively measured MVPA and gender (*p* < 0.001) and between LPA and age (*p* < 0.05). No other associations with objectively measured PA levels were observed for gender or age, nor for BMI, playground density, classroom density, or the presence of structured PA sessions.

## 4. Discussion

International recommendations for PA indicate that toddlers should have at least 180 min per day of PA at any intensity level [11]; however, there are limited accelerometry-based data that describe young children’s PA, in light of the new COVID-19 pandemic scenario. Thus, the aims of this study were two-fold: (a) to analyse toddlers’ PA levels and SB during school hours in ECEC institutions during the COVID-19 pandemic, as well as the adherence with specific recommendations on TPA and MVPA; and (b) to evaluate the child’s characteristics correlates (child’s age, gender, and BMI) of young children and school environment on toddlers’ TPA, light PA (LPA), MVPA, and SB during school hours in ECEC institutions.

The main findings indicate that throughout the school day, young children had a mean 154.35 ± 30.60 min of TPA, 111.85 ± 21.78 min of LPA, 42.50 ± 17.50 min of MVPA, and 164.47 ± 33.85 min of SB, with boys and girls exhibiting similar levels of TPA, LPA, and SB. In adherence to the Institute of Medicine recommendations, included toddlers exceeded the minimum one-quarter of time spent in the school facilities in TPA [13]. These results suggest that ECEC school hours provide the opportunity for an important contribution to the minimum recommendation of 3 h of daily TPA (85.75%, 82.81%, and 83.33% for boys and girls, respectively). Similar results have been reported elsewhere using pre-pandemic data. For instance, Zhang et al. [28] found that Canadian toddlers spent about 22 min/h in LPA, 6 min/h in MVPA, and 32 min/h in SB. In contrast, Ellis et al. [24] found that Australian children, aged 1 to 5 years, spent an average of 114 min in TPA during child care hours; that is, they were physically active for one-fifth of the time and spent around half their time sitting. In the same line, Zhang et al. [27] reported that young children may take part in more SB and less PA in ECEC centres over time. Thus, these results provide evidence that curricular practices aimed at toddlers, such as structured versus non-structured PA, indoor versus outdoor PA, and instructed versus non-instructed PA sessions, as well as the kind of curricular policy in the ECEC, could affect the amount of PA that they do during school hours.

The restrictions implemented in response to COVID-19 (e.g., social distancing regulations) had a negative impact on preschoolers’ PA levels [10,37,45]. These restrictions, in effect during the data collection phase in this study (October 2020 to May 2021), were strict for the entire population [46] and hit families in specific ways, for example, with the closure of public and private playgrounds. Consequently, ECEC institutions have assumed the responsibility to minimise, insofar as they can, the impacts of the pandemic on children’s PA levels [35]. Indeed, the results of this study suggest that participating ECEC institutions were aware of the importance of promoting PA within their school environment, in light of the public restrictions. Most young children in the sample met the international PA recommendations for spending a quarter of their school time being physically active [13] and fulfilled almost 85% of the WHO recommendations for TPA [11]. It is clear, then, that these ECEC institutions provide supportive environments for PA, each in their own way, especially during the COVID-19 era.

The COVID-19 pandemic could exacerbate the health effects of the obesogenic environment [47,48], so studying the relationship between BMI and PA/SB variables could provide insight into children’s health and education. According to the results of the current study, no association between BMI and LPA, MVPA, TPA, or SB was observed (Table 2). In the same line, studies on toddlers during the entire day, for instance, the one by Johansson et al. [17], in a sample of young children in Sweden, found that BMI was not correlated with 2-year- olds’ PA or SB. In the Netherlands, Wijtzes et al. [16] likewise failed to find associations between the BMI z-score and PA or SB. The present study’s results corroborate this evidence, showing that about half of the sample was categorised as normal weight, while the other half was evenly divided between underweight and overweight–obese. Notwithstanding, these results are alarming because almost 25% of the sample was overweight/obese. Using data collected prior to the pandemic, Cadenas-Sánchez et al. [49] showed that the prevalence of overweight and obese preschool children in Spain already ranged from 21.4% to 34.8%, and the pandemic seems to be causing an additional negative impact [47], creating problems for both the children’s present and future health [48]. The data reported here show an important prevalence of overweight and obesity in included toddlers, warranting additional efforts to promote a healthy weight in young children during this new pandemic time.

Children’s age was associated with LPA. Specifically, older children had approximately 16 min more LPA than younger ones. In the same line, using an observational system, Gubbels et al. [22] found that 3-year old children were more active outdoors during school hours than 2-year olds. Other studies, analysing PA during the entire day, have found similar associations. For example, Prioreschi et al. [21] found that TPA was higher in older age groups than younger ones, and Hager et al. [20] found that older toddlers engaged in nearly twice the amount of MVPA, compared to younger ones. Furthermore, Wijtzes et al. [16] found that levels of SB were lower and MVPA higher in older children. By contrast, Peden et al. [29] found that preschoolers were less likely to meet the Institute of Medicine [13] PA recommendation, compared to toddlers. The discordance between the results reported by Penden et al. [29] and the present study could be attributed to Peden et al.’s decision to compare PA between toddlers and preschoolers, not between young and older toddlers, as reported here. Furthermore, no association was observed between age and different intensities of PA (i.e., MVPA, TPA, and SB). Therefore, despite the finding that 24–36-month toddlers’ interest in continuous exploration grows as they do (as indicated by the increased LPA), more studies are necessary to explain how age could be a correlate of PA and SB during school hours.

Regarding toddlers’ gender and PA levels, boys exhibited higher levels of MVPA than girls (46.91 ± 19.63 min vs. 37.28 ± 12.92 min, respectively). This is in keeping with systematic review, showing a consistent association between gender and PA in young children [19,33]. The literature, focusing specifically on toddlers during school hours, reports similar results; for instance, Ellis et al. [24] found higher levels of MVPA in Australian boys, compared to girls. The role of gender in toddlers’ PA has been also evidenced during the ECEC time in the UK with higher levels of TPA and MVPA for boys [50,51]. Taken together, these findings suggest that PA measurements during ECEC hours might involve specific correlates (e.g., gender) that may not apply during the rest of waking hours (e.g., after school). In general, the difference observed between girls and boys in MVPA, which did not affect LPA, TPA, or SB, could be explained by the fact that teachers mediate the curricular practices related to MVPA in young children (i.e., structured or unstructured PA, free play, use of curricular activities in each type of activity), and these practices, especially the free play could promote more SB than MVPA in boys, compared to girls. However, future studies are needed, in order to examine how these curricular practices affect young children PA.

However, further studies are necessary to analyse specifically how MVPA is modulated by the different curricular practices (structured/unstructured PA opportunities, the role of the teacher, free/instructed play, among others) and by the location where these practices are performed (indoors/outdoors). In any case, following Lahuerta-Contell et al. [32], reaching the recommended levels of MVPA during school time requires further efforts to encourage less active children—with a particular eye on gender differences—to increase intensity during their curricular activities. For instance, Carson et al. [31] have found that sedentary time and MVPA in educators may be an important correlation of MVPA in toddlers during child care hours [31]. In this line, Tonge et al. [33] and Hnatiuk et al. [51] have argued that one way to reduce SB and promote PA for girls may be by having practitioners become actively involved with girls and encouraging girls to take part in MVPA from a very early age.

One of the novelties of this study relates to the exploration of the concordance between toddlers’ PA levels and ECEC classroom density. In line with its findings (i.e., an association between SB and classroom density), indoor environmental characteristics, such as the ‘modified open-plan space’, are positively associated with PA in toddlers, so providing enough spaces for playing likely keeps children engaged and active [27]. Indeed, a high density of preschoolers was associated with higher levels of SB during school hours [32]. Overall, it is possible that larger classrooms and/or lower student/teacher ratios could decrease SB and keep children active. According to the principals’ description of the impact of COVID-19 measures on classroom organization, the ratio of young children per classroom was not affected by the pandemic, suggesting that the association between the student/teacher ratio and PA could have more to do with the pedagogical and curricular practices with toddlers than with the pandemic regulations. Further research could elucidate how classroom density could affect SB, but in the meantime ECEC providers and ECEC educators should try to limit the use of equipment that restricts young children’s movement [13].

Regarding playground density, no association with PA levels was observed, which contradicts pre-pandemic studies in Dutch toddlers, wherein the size of the outdoor playground was positively correlated with children’s activity level during the outdoor time [22]. According to the descriptive data collected in the present study, even though some schools had a larger playground and a (non-significantly) lower density per square meter, children in different schools had similar levels of PA and SB. The dimensions of the playground environment showed that ECEC centres, with an outdoor environment measuring at least 400 m^2^, are associated with a 22% greater likelihood of meeting PA recommendations than schools with smaller playgrounds [30]; however, no ECEC institution involved in the current study had such a large playground. Research has shown that the pandemic has negatively impacted social patterns in children who typically engage with peers at school or through PA [36]. Furthermore, the restrictions implemented, including in ECEC institutions (e.g., limiting the mixing of classes), have something in common: the imposition of greater social distancing in and outside schools [52]. In fact, Lafave et al. [35] recently identified major barriers perceived by educators for promoting PA in the ECEC context during the pandemic, including limited PA space and access to equipment. Governmental regulations have mandated the staggering use of the playground during recess time to avoid crowding, guarantee social distancing, and prevent contact between different groups; this has been accomplished by distributing children into signposted sectors and reinforcing supervision during playtime [53]. Other vertical and horizontal contacts within school facilities have been forbidden to achieve the so-called ‘stable coexistence groups’ or ‘pods’, and this adaptation was very common among the participating ECEC institutions. Given that the daily schedule, in the sample classrooms, allocated time for outdoor activities and that all the schools had fixed equipment for PA in this outdoor space, the COVID-19 social distancing regulations for school playgrounds could be responsible for the positive PA levels in the present sample. In other words, shrinking the playground space and allowing fewer young children in the same space may have had a positive impact on PA. The present results suggest that having alternated breaks and specific time periods for playground use per classroom group, as well as increasing the use and enjoyment of the space by fewer children, may be positively responsible for schools meeting up to 85% of the full PA recommendations for this age group.

Given that most young children in this sample met recommendations (one-quarter of the time with active behaviours) and, in fact, almost completed the full 3 h of recommended PA per day during school hours, it is clear that these ECEC institutions provide, each with its own identity, and challenge the new COVID-19 scenario, with supportive environments for toddlers’ PA. However, taking into account that it is essential to examine curricular practices and environmental characteristics related to the school curriculum of ECEC centres relevant to children’s PA and SB while they are in the ECEC institutions, especially during the COVID-19 era, future studies analysing teachers’ perceptions of the barriers to PA that COVID-19 has created in schools are also needed.

### Strengths and Limitations

To our knowledge, this is the first study in the pandemic era (October 2020–May 2021) to examine toddlers’ PA levels and SB during ECEC time using objective measures with accelerometers. This is a novelty in COVID-19 studies, as previous studies have only described self-reported PA. A large sample of children across multiple ECEC institutions and classrooms was recruited, ensuring heterogeneity of centre characteristics.

On the other hand, this study is not without some limitations, including those inherent to its cross-sectional design and the lack of collecting socio-economic information from families. In addition, while accelerometry is an objective measure of PA and sedentary time, there is no consensus on the optimal cutoff in toddlers [39]. As other authors have suggested [31], it is possible that the choice of cutoffs used may have over- or underestimated the amount of sedentary time and PA in young children. Furthermore, the selection of the convenience sample does not allow generalisation of the results to the whole toddler population. In addition, this study was done in public ECEC institutions, excluding the private education sector. Future longitudinal studies may provide more information about the nature and direction of the associations between gender and objectively measured PA and SB using a stratified probabilistic sampling model involving a high diversity of ECEC institutions, because this would allow researchers and practitioners to have a global view in different contexts and increase the degree of generalisation of the impact of the pandemic on toddlers’ movement behaviour. In addition, future studies analysing the relationship between socio-economic variables and toddlers PA levels during all day hours are need.

## 5. Conclusions

The main findings of this study were that: (a) toddlers engaged in very high amounts of TPA and MVPA during ECEC hours, meeting 85% of the recommended levels of daily PA; (b) girls and boys displayed similar levels of LPA, TPA, and SB during school hours; (c) girls had lower levels of MVPA, compared to boys; (d) younger toddlers were less active than older ones; (e) BMI was not associated with PA of any intensity or SB; (f) playground and classroom density were not associated with higher levels of PA of any intensity, though classroom density was associated with SB. Taken together, 85% of the PA recommendations were met during school hours in the pandemic time. The present study shows that, in this age group (2–3 years), it is essential to focus on the analysis of total PA time; in other words, ECEC institutions and practitioners should encourage active behaviours, regardless of intensity. However, intervention studies aiming to promote PA at these ages should pay attention to the correlates of age and gender for each PA intensity.

## Figures and Tables

**Figure 1 children-09-00051-f001:**
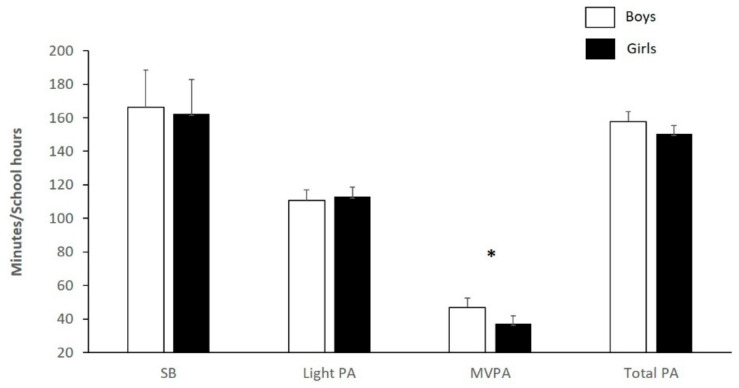
Physical activity and sedentary behaviour during school hours based on gender. SB, sedentary behaviour; light PA, light physical activity; MVPA, moderate to vigorous physical activity; total PA: light, MVPA. * *p* < 0.05 between boys and girls.

**Table 1 children-09-00051-t001:** Characteristics of toddlers (*n* = 120).

Variable	Measure
Age in months, mean (SD)	31.98 (3.162)
Gender, n (%)	
Boys	54.2%
Girls	45.8%
BMI, mean (SD) (age and sex adjusted)	16.04 (2.25)
Underweight (%)	23.3
Normal weight (%)	51.1
Overweight (%)	11.1
Obese (%)	14.4

SD: standard deviation.

**Table 2 children-09-00051-t002:** Mixed-model regressions for the association between explanatory variables and objective measures of physical activity.

	SB		LPA		MVPA		TPA	
Predictors	Estimates (95% CI)	*p*	Estimates (95% CI)	*p*	Estimates (95% CI)	*p*	Estimates (95% CI)	*p*
(Intercept)	226.93 (122.41, 331.44)	<0.001	132.71 (71.14, 194.28)	<0.001	46.16 (4.24, 88.08)	0.031	233.28 (143.53, 323.04)	<0.001
Gender (girl)	6.01 (−3.27, 15.28)	0.20	2.97 (−3.57, 9.51)	0.37	−7.97 (−13.48, −2.46)	0.005	−5.51 (−15.30, 4.29)	0.27
Age	12.57 (−7.05, 32.18)	0.21	−15.49 (−29.15, −1.84)	0.026	6.3 (−4.82, 17.42)	0.27	−8.92 (−29.30, 11.46)	0.39
Classroom density	−218.23 (−429.52, −6.93)	0.043	58.02 (−55.05, 171.10)	0.32	−59.22 (−122.66, 4.23)	0.067	−107.01 (−268.89, 54.88)	0.20
Playground density	−208.75 (−464.31, 46.82)	0.11	−43.84 (−179.05, 91.38)	0.53	29.47 (−44.32, 103.27)	0.43	−114.62 (−307.74, 78.51)	0.25
Random effects								
Total variance	571.1		285.75		205.54		640.74	
Random-effects variance	444.52		110.65		20.48		219.51	

## Data Availability

The data presented in this study are available on request from the corresponding author. The data are not publicly available, due to privacy or ethical restrictions.

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
