# Peer review of "Role of Spanish Toddlers’ Education and Care Institutions in Achieving Physical Activity Recommendations in the COVID-19 Era: A Cross-Sectional Study"

_children, 2022, doi:10.3390/children9010051_

Round 1

Reviewer 1 Report

  • The manuscript presents a relevant topic to publish in Children Journal, which could be accepted with some minor revisions.
  • The introduction provides adequate information and structure to set up the research questions raised in the manuscript; the methodology provides sufficient detail, but that can still be an improvement; results section is sufficiently clear and precise; the discussion of results based on previous literature; the conclusion of the study should be improved.
  • After carefully reading your manuscript, I point out some aspects that must be improved and corrected:

- The authors assume that the study has two objectives: (a) to analyse toddlers' PA levels and sedentary behavior (SB) during school hours in ECEC institutions, as well as the rate of adherence to specific recommendations on total PA (TPA) and moderate-vigorous PA (MVPA); and (b) to evaluate the sociodemographic correlates of young children and the school environment on toddlers' TPA, light PA (LPA), MVPA and SB during school hours in ECEC institutions." Regarding the second objective (b), the authors only explored two sociodemographic variables, specifically the child's age, and gender. It is too ambitious to put sociodemographic variables limited to the child's age and gender. I think that this objective should be reformulated, perhaps referring to the child's characteristics, because BMI was also explored in addition to age and gender.

  • In the abstract, the authors refer to the following: The main findings were that: (b) girls and boys displayed similar levels of active behaviors and SB, while girls had lower levels of MVPA compared to boys and younger toddlers were less active than older ones (lines 26-27); this finding is also written at the conclusion (lines 425-426); In my opinion, the phrase seems contradictory and should be rephrased. If girls have lower levels of MVPA compared to boys, then they do not have the same profile as boys in terms of active behaviors.

- The material and methods section should be revised. There are subtitles and text repeated throughout this manuscript section; please see what I marked and commented on in the manuscript. Authors should try to be more objective and consistent with the information written in each subchapter.

- the characterization of the sample should be more detailed, for example, the socio-economic status of the parents, the parents' academic qualifications. In fact, these variables should have been controlled in the statistical analysis.

- Statistical procedures should be better explained. Please, see my comments made on the manuscript;

- The strengths and limitations and conclusion sections should be revised as parts of the text are repeated.

- Some aspects of formatting should be corrected (spelling). Please, correct what is pointed out in the body of the manuscript;

- All statistical symbols must be in italics ( n, p, ....).

Author Response

Reviewer 1

Following the recommendations exposed by the reviewer 1 our itemised responses are listed below, while our modifications to the text have been written in red throughout the manuscript to make its revision easier for the reviewer. We believe we have thoroughly addressed all reviewer concerns, and we appreciate their taking the time and energy to help us improve the paper.

  • The authors assume that the study has two objectives: (a) to analyse toddlers' PA levels and sedentary behavior (SB) during school hours in ECEC institutions, as well as the rate of adherence to specific recommendations on total PA (TPA) and moderate-vigorous PA (MVPA); and (b) to evaluate the sociodemographic correlates of young children and the school environment on toddlers' TPA, light PA (LPA), MVPA and SB during school hours in ECEC institutions." Regarding the second objective (b), the authors only explored two sociodemographic variables, specifically the child's age, and gender. It is too ambitious to put sociodemographic variables limited to the child's age and gender. I think that this objective should be reformulated, perhaps referring to the child's characteristics, because BMI was also explored in addition to age and gender.

Answer: Thank you. We have reformulated accordingly. Please see the new version of the manuscript (see lines 22-24 and 100-102).

“(b) to evaluate the child's characteristics correlates (child's age, gender and BMI) of young children and the school environment on toddlers' TPA, light PA (LPA), MVPA and SB during school hours in ECEC institutions”.

  • In the abstract, the authors refer to the following: The main findings were that: (b) girls and boys displayed similar levels of active behaviors and SB, while girls had lower levels of MVPA compared to boys and younger toddlers were less active than older ones (lines 26-27); this finding is also written at the conclusion (lines 425-426); In my opinion, the phrase seems contradictory and should be rephrased. If girls have lower levels of MVPA compared to boys, then they do not have the same profile as boys in terms of active behaviors.

Answer: Thank the reviewer for these comments. We are sorry for not being clear enough. This sentence has been reformulated accordingly (see lines 25-27 and 413, respectively).

“(b) girls and boys displayed similar levels of LPA, TPA and SB, while girls had lower levels of MVPA compared to boys and younger toddlers were less active than older ones”.

“(b) girls and boys displayed similar levels of LPA, TPA and SB during school hours”

  • The material and methods section should be revised. There are subtitles and text repeated throughout this manuscript section; please see what I marked and commented on in the manuscript. Authors should try to be more objective and consistent with the information written in each subchapter.

Answer: Thanks for the comment.

Paragraph modified:

(see page 4, lines 146-147).

“All ECECs were located in socioeconomically similar areas”. 

Answer: Thanks for the comment. Following the reviewer’s recommendation we have improved the methods section by describing rules and regulations derived from COVID under a new sub-title (lines 160-170).

Paragraph modified:

 “2.3. ECEC and COVID-19 regulations

The rules of social distancing, hygiene, and cleanliness, as well as other logistical and institutional rules for both teachers and children remained in place from the beginning of the school year (September 2020) to the end (June 2021). These policies were common for all the ECEC institutions. Following governmental regulations for maintaining a safe distance and avoiding contact between different groups, one of the restrictions imposed to prevent the spread of COVID was the creation of stable classroom groups that do not socialise with children outside their group. Furthermore, school principals were asked how the different groups were organised during recess time to comply with the new COVID-19 regulations, including what modifications had been made to the play area and how many children per space participated during playtime”.

We think the content in the current version is more objective and coherent than in the previous one. In addition, following the reviewer’s advice we have moved some paragraphs found in the description of the EECs in order to explain better the methodological approach, checking subheadings, etc. …

(lines 176-177)

“The ECEC teachers responsible for each classroom placed the accelerometer on the children at the beginning of the school day (9:00-9:15 am) and removed it a few minutes before the end (4:15-4:30 pm)”.  

Furthermore, following the reviewer’s suggestion we have moved some information (in the results section) to the methodological part (e.g. when describing the educational contexts)

(see page 4, lines 156-159).

“Of the seven participating ECEC institutions, three had specific structured PA sessions within their curricular practices (in the weekly schedule and in a special classroom), while the other four worked PA curricular contents in an integrated, interdisciplinary way . All ECECs had fixed equipment in their outdoor playgrounds”.

  • the characterization of the sample should be more detailed, for example, the socio-economic status of the parents, the parents' academic qualifications. In fact, these variables should have been controlled in the statistical analysis.

Answer: We appreciate the reviewer’s comment. At the same time, we apologize for not being clear enough in the use of some terms included in the submitted version. Please, note that in our original questionnaire “family status” referred to the children’s care givers condition at home (i.e, living with parents, living with divorced parents or living with adults different to parents. That is, we referred to “family status” as the way to get information identifying who lives with the children at home, who goes with her/him to the school, etc. However, in order to be clear, in the revised version of the manuscript we have deleted the term “family status” because even though it was included in the analyses originally, the association of “family status” was not significant and was not included in the final models. Taking into account that our research is focused on toddlers’ movement behaviour during school hours, in our questionnaire we did not collect the family’s socio-economic status and/or the parents' academic qualifications because our purposes were related to the school setting. In our study the way used to control any related effect of socio-economic status was that all ECECs were located in socioeconomically similar areas. Furthermore, we have added a new paragraph in the limitation section explaining that for future studies analysing all day hours it will be necessary to explore the association between these socio-economic variables and the toddlers’ physical activity levels.

New paragraphs in the limitation section:

“On the other hand, this study is not without some limitations, including those inherent to its cross-sectional design and the lack of collecting socio-economic information from families”.

“In addition, future studies analysing the relationship between socio-economic varia-bles and toddlers PA levels during all day hours are need”.

  • Statistical procedures should be better explained. Please, see my comments made on the manuscript;

Answer: We appreciate comment. According to the reviewer’s suggestion, we have removed the statement related to inferences because we actually have realized we did not use this method in the final stage of statistical analysis: "Finally, one-way analysis of variance and the Bonferroni correction were used to estimate differences between girls and boys on LPA, MVPA, TPA and SB". For inference, we relay in the effect estimates of the linear mixed effects models, providing corrected standard errors and p-value for the null hypothesis of equal average effects of sex on LPA, MVPA, TPA and SB. We employed linear mixed models because the correlated nature of the children measures within classrooms and schools might suggest the use of statistical tools such as the t-test designed for independent samples although useful for the research problem at hand; it could be replaced by the linear mixed methodology, which considers the different sources of variation such as the classroom level and the school level, adjusting the standard errors for the inference, as it is shown by Galecki y Burzikowski (2013).

Galecki, A. & Burzikowski, T. (2013). Linear Mixed Effects Models using R: a step by step approach. New York, NY: Springer.

  • The strengths and limitations and conclusion sections should be revised as parts of the text are repeated.

Answer: We appreciate the comment. Accordingly, the limitation section has been modified. We have deleted and moved some parts in the discussion and conclusion sections in order to be clearer enough (see lines 379-387, 395-396 and 407-309). Furthermore, the Conclusion section has been shortened and, in our opinion, it is currently more objective and non-repetitive than the original version.

New lines in the limitation section:

“On the other hand, this study is not without some limitations, including those inherent to its cross-sectional design and the lack of collecting socio-economic information from families”.

“In addition, future studies analysing the relationship between socio-economic variables and toddlers PA levels during all day hours are need”.

We also have moved one paragraph from the Conclusion section to the end of the Discussion section:

“Given that most young children in this sample met recommendations (one-quarter of the time with active behaviours) and in fact almost completed the full 3 hours of recommended PA per day during school hours, it is clear that these ECEC institutions provide, each with its own identity, and challenging the new COVID-19 scenario, supportive environments for toddlers' PA. However, taking into account that it is essential to examine curricular practices and environmental characteristics related with the school curriculum of ECEC centres relevant to children’s PA and SB while they are in the ECEC institutions, especially during the COVID-19 era, future studies analysing teachers’ perceptions of the barriers to PA that COVID-19 has created in schools are also needed”.

  • Some aspects of formatting should be corrected (spelling). Please, correct what is pointed out in the body of the manuscript;

Answer: Amended.

  • All statistical symbols must be in italics ( n, p, ....).

Answer: Amended.

Somme comments in the text:

  • I don't agree with your idea! Are you suggesting that boys are more stimulated by the teacher?

Answer: Thank the reviewer for this comment. We apologize for the misunderstanding. It is worth to note that what we suggest from the results found related to the effect of gender in physical activity in favour of boys (see page lines 209-214 and Figure 1) is the possibility that, in terms of movement behaviour, structured activity and no free play could led young girls to be more physically active. In other words, the free play for girls could be related more with sedentary behaviours; therefore, the active role of teacher could counterbalance this sedentary behaviour by encouraging girls to be physically active. We have incorporated a new paragraph as follows (see lines 316-320):

“In general, the difference observed between girls and boys in MVPA, which did not affect LPA, TPA or SB, could be explained by the fact that teachers mediate the curricular practices related to MVPA in young children (structured or unstructured PA, free play, use of curricular activities in each type of activity), and these practices, especially the free play could promote more SB than MVPA in boys compared to girls. However, future studies are need in order to examine how these curricular practices affect young children PA”.

Reviewer 2 Report

First of all, I would like to thank the authors for their contribution.

The introduction must be written in a clearer and more rigorous manner.

The paper is well-positioned within the literature and the references are adequate.

The data analysis was done meticulously and clearly presented in the form of tables.

Author(s) need to mention ethical issues for their study. I propose to add the following reference:

Petousi, V., & Sifaki, E. (2020). Contextualizing harm in the framework of research misconduct. Findings from discourse analysis of scientific publications, International Journal of Sustainable Development, 23(3/4), 149-174, DOI: 10.1504/IJSD.2020.10037655

https://www.inderscienceonline.com/doi/abs/10.1504/IJSD.2020.115206

In sum, I applaud all the efforts of the author(s) for this research.

Author Response

Reviewer 2

Following the recommendations exposed by the reviewer 2 our itemised responses are listed below, while our modifications to the text have been written in red throughout the manuscript to make its revision easier for the reviewer. We believe we have thoroughly addressed all reviewer concerns, and we appreciate their taking the time and energy to help us improve the paper.

  • The introduction must be written in a clearer and more rigorous manner.

Answer: We thank the reviewer for this comment. We have checked the Introduction and we have incorporated a new paragraph.

New paragraph:

“By the other hand, the pandemic has driven schools to make adaptations to their internal operations, creating measures for prevention, hygiene and health promotion in the face of COVID-19 [5]. This new scenario has impacted movement behaviours for children and entailed new challenges for teachers, including the improvement of methodological competences related to conducting PA classes that are attractive and adapted to the conditions during the pandemic, as well as training educators to motivate pupils to be more physically active [34,35]”.

  • The paper is well-positioned within the literature and the references are adequate.
  • The data analysis was done meticulously and clearly presented in the form of tables.

Answer: We appreciate the reviewer’s positive feedback.

  • Author(s) need to mention ethical issues for their study. I propose to add the following reference:

Petousi, V., & Sifaki, E. (2020). Contextualizing harm in the framework of research misconduct. Findings from discourse analysis of scientific publications, International Journal of Sustainable Development, 23(3/4), 149-174, DOI: 10.1504/IJSD.2020.10037655

https://www.inderscienceonline.com/doi/abs/10.1504/IJSD.2020.115206

Answer: We thank the reviewer for sharing this manuscript. After reading the paper by Peousi and Sifaki (2020) carefully, we are glad to confirm that in our study we have following the mandatory ethical standards such as previous ethical committee approval, parents’ and teachers’ written consent, etc. The study was conducted according to the guidelines of the Declaration of Helsinki, and approved by the Ethics Committee of University of Valencia (Ethical approval code-UV-INV_ETICA-1441131). Furthermore, we have added the reference.

  • In sum, I applaud all the efforts of the author(s) for this research.

Answer: We appreciate the reviewer’s positive feedback.